# DDC-Promoter-Driven Chemogenetic Activation of SNpc Dopaminergic Neurons Alleviates Parkinsonian Motor Symptoms

**DOI:** 10.3390/ijms24032491

**Published:** 2023-01-27

**Authors:** Dong-Chan Seo, Yeon Ha Ju, Jin-Ju Seo, Soo-Jin Oh, C. Justin Lee, Seung Eun Lee, Min-Ho Nam

**Affiliations:** 1Research Animal Resource Center, Korea Institute of Science and Technology (KIST), Seoul 02456, Republic of Korea; 2Brain Science Institute, Korea Institute of Science and Technology (KIST), Seoul 02456, Republic of Korea; 3Department of Integrated Biomedical and Life Sciences, College of Health Science, Korea University, Seoul 02841, Republic of Korea; 4Technological Convergence Center, Korea Institute of Science and Technology (KIST), Seoul 02456, Republic of Korea; 5Center for Cognition and Sociality, Institute for Basic Science, Daejeon 34126, Republic of Korea; 6Department of KHU-KIST Convergence Science & Technology, Kyung Hee University, Seoul 02447, Republic of Korea

**Keywords:** Parkinson’s disease, DOPA decarboxylase, promoter, chemogenetics

## Abstract

Parkinson’s disease (PD) is a neurodegenerative disorder with typical motor symptoms. Recent studies have suggested that excessive GABA from reactive astrocytes tonically inhibits dopaminergic neurons and reduces the expression of tyrosine hydroxylase (TH), the key dopamine-synthesizing enzyme, in the substantia nigra pars compacta (SNpc). However, the expression of DOPA decarboxylase (DDC), another dopamine-synthesizing enzyme, is relatively spared, raising a possibility that the live but non-functional TH-negative/DDC-positive neurons could be the therapeutic target for rescuing PD motor symptoms. However, due to the absence of a validated DDC-specific promoter, manipulating DDC-positive neuronal activity has not been tested as a therapeutic strategy for PD. Here, we developed an AAV vector expressing mCherry under rat DDC promoter (AAV-rDDC-mCherry) and validated the specificity in the rat SNpc. Modifying this vector, we expressed hM3Dq (Gq-DREADD) under DDC promoter in the SNpc and ex vivo electrophysiologically validated the functionality. In the A53T-mutated alpha-synuclein overexpression model of PD, the chemogenetic activation of DDC-positive neurons in the SNpc significantly alleviated the parkinsonian motor symptoms and rescued the nigrostriatal TH expression. Altogether, our DDC-promoter will allow dopaminergic neuron-specific gene delivery in rodents. Furthermore, we propose that the activation of dormant dopaminergic neurons could be a potential therapeutic strategy for PD.

## 1. Introduction

Parkinson’s disease (PD) is the second most common neurodegenerative disorder which causes typical motor symptoms, such as resting tremor, rigidity, bradykinesia, and postural instability [1,2]. These parkinsonian motor symptoms have been known to be brought about by dopamine (DA) depletion in the nigrostriatal pathway [3]. The DA depletion has long been mainly attributed to extensive DA neuronal death in substantia nigra pars compacta (SNpc). Due to the long belief in irreversible neuronal death, no disease-modifying therapy has been discovered [4]. A number of previous studies have investigated the mechanism underlying DA neuronal death. For instance, dysfunctional autophagy-lysosome system [5], synucleinopathy-associated endoplasmic reticulum (ER) stress [6,7,8], disturbance in mitochondrial function [9], and dysregulated calcium homeostasis [10,11] have been considered to contribute to the DA neuronal death. However, whether and how DA depletion could be attributed to the functional suppression of DA neurons rather than their extensive death has been less highlighted.

Meanwhile, we recently demonstrated that reactive astrocytes aberrantly synthesize inhibitory transmitter GABA through the enzymatic action of MAO-B, tonically inhibiting neighboring DA neurons in the SNpc of the brains of PD animal models and patients [12,13,14,15]. Particularly, astrocytic GABA-mediated tonic inhibition of DA neuronal firing downregulates the expression of tyrosine hydroxylase (TH), the key DA-synthesizing enzyme, in the SNpc neurons. On the other hand, the expression of another DA-synthesizing enzyme, DOPA decarboxylase (DDC) is relatively spared. Moreover, the existence of the distinct population of remaining, but non-functional, DA neurons which express DDC, but not TH, called dormant DA neurons, has been reported [12]. In this regard, manipulating the activity of dormant DA neurons could be a potential therapeutic strategy for restoring DA synthesis and release in the nigrostriatal pathway of the PD brains.

In spite of the potential importance of DDC-positive dormant DA neurons, it has been impossible to specifically manipulate the DDC-positive dormant DA neuronal activity because DDC-specific promoter has been lacking. Moreover, other dopaminergic neuron-specific genes, such as TH and dopamine transporter (DAT), have not been widely utilized as cell-type-specific promoters. The reason why optimization of a cell-type-specific promoter in an adeno-associated virus (AAV) vector is challenging is because the in vivo specificity is not fair enough in most cases [16,17]. So far, only a few genes, such as human-synapsin (hSynapsin) and glial fibrillary acidic protein (GFAP), have been developed as cell-type-specific promoters in the viral vectors.

Here, we firstly developed and characterized the rat DDC (rDDC)-specific promoter, which shows a fair specificity to DDC^+^ neurons in the rat SNpc. Furthermore, we demonstrated that our DDC-promoter-driven chemogenetic activation of SNpc dopaminergic neurons significantly alleviated the nigrostriatal TH expression and the parkinsonian motor symptoms in a rat model of PD. Given these findings, we suggest our DDC-promoter as a valid tool for targeting dormant DA neurons. We also propose that the activation of DDC-positive dormant DA neurons could be an innovative strategy for PD therapy.

## 2. Results

### 2.1. Design and Specificity Validation of the DDC-Promoter in an AAV Vector

To cell-type-specifically manipulate the gene expressions using AAV, one of the most widely used methods as a foreign gene carrier in the brain, the specific promoter is necessary. We first designed rDDC-promoter on an approximately 1.6 kb region upstream from the transcription start site (TSS) in the DDC gene sequence of *Rattus norvegicus*, considering the small packaging capacity of AAV limited to ~4.7 kb.

To validate the specificity of the promoter in an AAV vector, we constructed the AAV vector to express fluorescent reporter protein mCherry under the DDC promoter (AAV-rDDC-mCherry) (Figure 1a). The packaged AAV virus was injected into the rat SNpc using stereotaxic apparatus because SNpc is one of the core regions of DA neurons (Figure 1b). After two weeks from the virus injection, we performed immunostaining to assess the specificity of the DDC-promoter (Figure 1c). We found that 73.9% of mCherry-expressing neurons driven by DDC-promoter were DDC-immunoreactive (Figure 1d,e). These results indicate that the promoter shows a fair DA neuron-specificity in SNpc.

### 2.2. Chemogenetic Activation of DDC^+^ Neurons Alleviates PD Motor Symptoms

Next, to manipulate the DDC^+^ neuronal activity using a chemogenetic approach, we inserted the hM3Dq gene in the multi-cloning site (MCS) AAV vector with rDDC-promoter (Figure 2a). hM3Dq is a Gq-protein coupled designer receptor exclusively activated by designer drug (DREADD), which is broadly utilized for stimulating neuronal activity [18]. We first tested whether hM3Dq can indeed enhance the neuronal activity upon clozapine N-oxide (CNO) administration. We injected the AAV-rDDC-hM3Dq-mCherry virus into rat SNpc (Figure 2b). Two weeks later, we performed loose-cell attached patch with the virus-infected neurons to record the spontaneous firing (Figure 2c,d). We found that the mean firing rate was significantly increased by ~65% upon bath application of CNO (5 μM) (Figure 2e,f). This result indicates that the DDC^+^ neuron-specific activation of hM3Dq faithfully boosts the neuronal firing.

We previously reported that channelrhodopsin-mediated optogenetic activation of remaining SNpc neurons in the PD rodent models partially, but significantly, rescued the PD motor symptoms [12]. However, because the channelrhodopsin was non-specifically expressed in the SNpc neurons by controlling pan-neuronal promoter hSynapsin, it was hard to conclude whether manipulating the dormant DA neuronal activity is indeed responsible for the efficacy. To test whether the DDC-promoter-driven hM3Dq-mediated activation of the dormant neurons could rescue the PD motor symptoms and pathology in vivo, we utilized the rat PD model of viral A53T-mutated alpha-synuclein overexpression (A53T PD model) [12,19,20]. This model is known to exhibit substantial damage to nigrostriatal DA neurons and PD motor symptoms within three weeks [12].

To specifically express hM3Dq in the dormant DDC^+^ neurons, we injected the AAV-rDDC-hM3Dq-mCherry into the SNpc. AAV-rDDC-mCherry was used as the control. Two weeks later, we injected the AAV-CMV-A53T virus to induce PD pathology. After three weeks from the AAV-CMV-A53T virus injection, when a distinct population of TH^−^/DDC^+^ dormant neurons is reported to be present in the SNpc [12], we intraperitoneally administered CNO (1.5 mg/kg) twice for all rats with an 18-h interval (Figure 3a). To assess the motor function, we conducted two behavioral tests, the stepping test [21] (Figure 3b) and the balance beam test [22] (Figure 4a) at three different time points: before AAV-CMV-A53T injection (pre), before CNO treatment (PD), and after CNO treatment (post). From the stepping test, we found that the contralateral forepaw showed a significant deficit in adjusting steps due to A53T-alpha-synuclein overexpression, indicating PD-like motor symptoms. Intriguingly, we found that CNO treatment significantly recovered the stepping ratio of the impaired forepaw in the hM3Dq group, but not in the control group (Figure 3c). Likewise, we found that the motor coordination assessed by the balance beam test was significantly impaired by A53T-alpha-synuclein overexpression, evidenced by increased foot slip numbers and reduced balance beam scores. Consistently, CNO treatment significantly reduced the foot slip numbers and improved balance beam scores to the normal level in the hM3Dq group, but not in the control group (Figure 4b,c). These findings suggest that chemogenetic activation of SNpc DDC^+^ neurons significantly alleviates the PD-like motor symptoms in the A53T PD model.

### 2.3. Chemogenetic Activation of DDC^+^ Neurons Rescues Nigrostriatal TH Loss

Next, we sought to investigate whether DDC-promoter-driven chemogenetic activation of SNpc DDC^+^ neurons indeed activated the neuronal activity. Thus, we assessed the expression of c-Fos, an immediate early gene, by performing immunohistochemistry (Figure 5a). We found that hM3Dq-mediated chemogenetic activation significantly increased the portion of c-Fos+ neurons out of hM3Dq-mCherry-expressing neurons (Figure 5b).

We subsequently tested if this chemogenetic activation rescues nigrostriatal TH loss in accordance with the behavioral recovery. We performed immunohistochemistry with anti-TH antibody and counted the number of TH^+^ neurons (Figure 6a). We found that A53T-mutated alpha-synuclein overexpression induced a significant loss of TH throughout the whole SNpc (Figure 6). We also found that CNO treatment significantly increased the number of remaining TH^+^ neurons in the ipsilateral SNpc of the hM3Dq group rats (Figure 6). This finding implicates that chemogenetic activation of SNpc DDC^+^ neurons could awaken the dormant DA neurons to regain the TH expression.

Based on the fact that SNpc DA neurons project to the striatum (called nigrostriatal pathway) [23], we further evaluated the striatal TH density (Figure 7a). We found that the A53T-mutated alpha-synuclein overexpression decreased the TH expression by 49% in the ipsilateral striatum, compared to the contralateral side. On the other hand, CNO-mediated hM3Dq-activation of DDC^+^ neurons significantly restored the TH expression in the ipsilateral striatum to the level of 80% compared to the contralateral side (Figure 7b). These findings indicate that DDC-specific promoter-driven chemogenetic activation of SNpc DDC^+^ neurons rescues the nigrostriatal TH expression in the DA neurons, which could lead to the recovery of PD-like motor symptoms.

## 3. Discussion

In this study, we constructed and characterized the DDC-specific promoter in an AAV vector to investigate the degree of DA neuron-specific expression in the rat brain. Using this promoter, we specifically targeted the dormant DA neurons in a rat PD model. We also tested the therapeutic efficacy of hM3Dq-mediated chemogenetic activation of DDC^+^ neurons using the AAV gene delivery system for rescuing the nigrostriatal TH expression and alleviating the motor symptoms.

### 3.1. The Necessity of DDC Promoter Validation

PD is well documented as a neurodegenerative disorder characterized by an extensive loss of dopaminergic neurons in the SNpc. However, we have recently demonstrated that there is a distinct population of non-functional, but still, live, DA neurons in the SNpc of PD brains [12]. We further demonstrated that these non-functional DA neurons lose TH expression first, while their DDC expression is relatively spared. They could cell-autonomously regain their TH expression by optogenetic activation. Therefore, we named them “dormant” DA neurons. Previous studies have also strongly supported the idea that TH expression is regulated upon neuronal activity [24,25,26]. More importantly, the surviving neurons could be the potential therapeutic target of PD. Indeed, the marked efficacy of levodopa, which needs to be converted to DA via the enzymatic action of DDC, could be attributed to the presence of the dormant DDC^+^ neurons. Therefore, to develop novel gene therapy for treating PD, it could be important to specifically target the remaining DDC^+^ neurons in the SNpc. Unfortunately, there has been no such validated promoter specifically targeting this remaining population. We, here, focused on DDC, which has been merely noticed as a promoter target, unlike TH or DAT. A DDC-specific promoter could be utilized to specifically target both healthy DA neurons in the physiological condition and dormant DA neurons in the PD condition.

### 3.2. Neuromodulation Strategy for PD Treatment

Neuromodulation using chemogenetic or optogenetic methods has been utilized for alleviating the motor symptoms of PD. In particular, neurons in the subthalamic nucleus [21], glutamatergic neurons in the cuneiform nucleus [27], GABAergic neurons in the entopeduncular nucleus [28], and medium spiny neurons in the striatum [29,30,31] have been highlighted as the target neurons for optogenetic inhibition or activation. In detail, we previously reported that optogenetic inhibition of GABAergic neurons in the entopeduncular nucleus (homologous to the globus pallidus in primates) significantly alleviated the akinesia in 6-OHDA rat models [28]. Furthermore, the optogenetic inhibition of subthalamic nucleus neurons was also reported to significantly reduce the motor dysfunction in the same model [21]. In addition to the optogenetic manipulation of the neuronal activity, chemogenetic manipulation of specific neuronal populations in the pedunculopontine area [32], motor cortex [33], and subthalamic nucleus [34] have been considered to be utilized for rescuing the PD symptoms in animals. Particularly, chemogenetic activation of striatal cholinergic neurons alleviated parkinsonian motor symptoms [35], while chemogenetic inhibition of striatal projection neurons rescued the symptoms [36]. Even though the previous reports have accentuated the therapeutic potential of neuromodulation in PD [12,27,29,30,37,38,39], they have not tested whether chemogenetic activation of the remaining DAergic neurons in the SNpc could be effective for alleviating PD motor symptoms. In the current study, we utilized the newly validated DDC promoter for cell-type-specifically manipulating the remaining DA neurons in the SNpc, and we demonstrated that the chemogenetic activation of DDC^+^ neurons significantly alleviated the motor symptoms and nigrostriatal TH loss.

### 3.3. Advantages of AAV-Mediated Gene Therapy

Our study utilized AAV for gene delivery to the SNpc neurons. Viral vectors are frequently utilized for gene delivery because they offer several advantages: effectiveness, stability, and long-lasting protein expression [40,41,42]. Especially, AAV has been most broadly employed due to its low immunogenicity and cytotoxicity [41,43]. Indeed, natural infections with wild-type AAV are known to be generally harmless. Even after AAV gene therapy, AAV mostly remains episomal, with a low proclivity to integrate into the genome. Based on the safety, in 2017, Luxturna, the very first AAV as a drug, was approved by the US Food and Drug Administration (FDA) [44]. Since then, several AAV drugs have been approved and applied for human use, for instance, ABECMA, ADSTILADRIN, ALLOCORD, and so on [45]. In addition, to improve the efficacy and safety of gene therapy, cell-type-specific promoter validation is important. If the gene is delivered and expressed in undesirable cell populations, severe negative consequences may occur [46]. Therefore, the AAV-mediated DA-neuron targeted delivery of a chemogene driven by the DDC-specific promoter would be applicable to human patients in the future.

### 3.4. Limitation of Chemogenetic Manipulation as a Therapeutic Strategy

For chemogenetic manipulation, we utilized hM3Dq as an excitatory DREADD and CNO, consistent with previous studies [12,27,29,30,37,38,39]. Neurons expressing hM3Dq show immediate depolarization and activation upon CNO treatment through well-documented Gq-mediated downstream signaling [47]. In an in vivo condition, CNO is known to show efficacy within several minutes, which lasts at least for several hours following intraperitoneal administration [47]. This could be why our short-term CNO administration significantly recovers the motor function and TH expression in the PD model rats. When the CNO administration is ceased, the DA neurons are expected to be returned to the original suppressed state in a few days so that PD-like motor symptoms could be demonstrated again. Therefore, for long-lasting effects, the repeated administration of CNO might be needed.

## 4. Materials and Methods

### 4.1. Vector Design and Virus Production

The rDDC gene locus was identified using the UCSC Genome Browser on Rat Jul. 2014 (RGSC 6.0/rn6) Assembly. A 1.6-kb upstream putative promoter region of the DDC located on the genomic sequence from Chromosome14q21 (chr14-:91,998,135–91,996,504). rDDC-promoter was generated into MluI-BamHI restriction enzyme sites of AAV-MCS (multi-cloning site) expression vector (Cell Biolabs, Inc., cat# VPK-410, California, USA) by in-Fusion cloning scheme as described previously [48]. The AAV-rDDC-mCherry and AAV-rDDC-hM3Dq-mCherry plasmids in which mCherry and hM3Dq-mCherry gene expression are regulated by the 1.6 kb promoter were inserted into the BamHI-BsrGI restriction site of the AAV-rDDC-MCS vector and verified by sequencing.

The viral vectors were pseudotyped, where the transgene of interest was flanked by inverted terminal repeats of the AAV2 packaged in an AAV-DJ capsid. AAV-DJ was engineered via DNA family shuffling technology, which created a hybrid capsid from 8 AAV serotype. AAV vectors were thereafter purified by iodixanol gradients by the KIST Virus Facility. The production titers were 2.66 × 10^13^ genome copies/mL (GC/mL) and 1.27 × 10^13^ for AAV-rDDC-mCherry and AAV-rDDC-hM3Dq-mCherry, respectively. Additionally, the titer of AAV-CMV-A53T for the A53T PD model was 4.54 × 10^13^ GC/mL.

### 4.2. Animals

Twenty male Sprague-Dawley rats (7- to 8-weeks-old) were obtained from Daehan BioLink, Eumseong-gun, Korea. All rats were in a constant temperature- and humidity-controlled environment with a 12-h light-dark cycle (lights on at 8 am) and food and water were freely fed. The rats, including the A53T PD model and control, male SD rats (9-weeks-old when injecting AAV-CMV-A53T) were used.

#### A53T PD Model Preparation

Stereotaxic administration of AAV-CMV-A53T virus (Right rat SN: AP -−5.6 mm, ML −2.0 mm relative to the bregma, DV −7.5 mm relative to the dura; 0.2 μL/min, 1:1 diluted with saline, total 2 μL), was performed under respiratory anesthesia induced by isoflurane.

### 4.3. Chemogenetics

Eight rats for each group were unilaterally injected with AAV-rDDC-mCherry or AAV-rDDC-hM3Dq-mCherry into the right SNpc. Two weeks after viral injection, all of the rats injected AAV-CMV-A53T into the same region. Three weeks after, the rats were intraperitoneally administrated with CNO (1.5 mg/kg, at 7 pm of day 22 and 1 pm of day 23) and underwent the behavioral assays.

### 4.4. Spontaneous Action Potential Firing Recording

Rats were deeply anesthetized using vaporized isoflurane and then decapitated. After decapitation, the brain was quickly excised from the skull and submerged in ice-cold cutting solution (250 mM of sucrose; 26 mM of NaHCO3; 10 mM of d-(+)-glucose; 4 mM of MgCl2; 3 mM of myo-inositol; 2.5 mM of KCl; 2 mM of sodium pyruvate; 1.25 mM of NaH2PO4; 0.5 mM of ascorbic acid; 0.1 mM of CaCl2; and 1 mM of kynurenic acid, pH 7.4). Horizontal slices (300 μm thick) of SNpc were prepared with using a vibrating microtome (D.S.K Linear Slicer, Kyoto, Japan). For stabilization, slices were incubated at room temperature for at least 1 h in a solution containing 130 mM of NaCl; 3.5 mM of KCl; 24 mM of NaHCO3; 1.25 mM of NaH2PO4; 1.5 mM of CaCl2; 1.5 mM of MgCl2; and 10 mM of d-(+)-glucose, pH 7.4.

For spontaneous action potential firing recording, the current was recorded under oxygenated aCSF solution by cell-attached voltage-clamp. The recording electrode (5–8 MΩ) was fabricated standard-wall borosilicate glass (Warner Instrument Corp., MA, USA) and filled with a KCl-based internal solution (150 mM of KCl; 10 mM of HEPES; 1 mM of CaCl2 and 1 mM of MgCl2; with pH adjusted to 7.3 and osmolality adjusted to 292 mOsmol/kg). Electrical signals were digitized and sampled at 10 ms intervals with Digidata 1550 data acquisition system and the Multiclamp 700B Amplifier (Molecular Devices, CA, USA). The frequency of spontaneous action potential before and after Clozapine N-Oxide (CNO) (Tocris, #4936, Bristol, UK) 5 uM administration was analyzed by the pClamp10.2 software.

### 4.5. Immunofluorescence Staining and Confocal Imaging

Cardiac perfusion was performed under deep anesthesia using 2% avertin with normal saline and ice-chilled 4% PFA solution immediately after the last behavior test finished (1.5 to 2 h after the last CNO administration). After cardiac perfusion, the brains were stored in 4% PFA solution at 4 °C overnight to post-fixation and then stored in 30% sucrose solution for more than one day at 4 °C for cryo-protection. The 30-μm-thick coronal SNpc sections were used for immunofluorescence staining. The slices were incubated for 2 h in a blocking solution (0.3% Triton-X100 and 2% normal serum in 0.1 M PBS) and in a mixture of primary antibodies in a blocking solution for 16 h at 4 °C after blocking. Primary antibodies used in this study are chicken anti-tyrosine hydroxylase (diluted in 1:500; Abcam, ab76442, Cambridge, UK), rabbit anti-DOPA decarboxylase (diluted in 1:500; Abcam, ab3905, Cambridge, UK), guinea pig anti-c-fos (diluted in 1:500; Synaptic systems, 226 004, Goettingen, Germany), and mouse anti-alpha synuclein (diluted in 1:500; Abcam, ab1903, Cambridge, UK). Then, the sections were washed with PBS three times and incubated with fluorescent secondary antibodies for 2 h at room temperature. Secondary antibodies used in this study are donkey anti-chicken 488 (diluted in 1:500; Jackson, 703-545-155, ME, USA), donkey anti-mouse IgG Alexa 647 (diluted in 1:500; Jackson, 715-605-150, ME, USA), and donkey anti-rabbit Alexa 647 (diluted in 1:500; Jackson, 711-605-152, ME, USA), and washed with PBS three times. Optionally, DAPI (diluted in 1:3000; Pierce, D1306, MA, USA) staining was performed. Lastly, the samples were mounted with a fluorescent mounting medium (Dako, S3023, Hovedstaden, Denmark) and dried. The images of these samples were obtained with a confocal microscope (Nikon, A1, Tokyo, Japan), and Z stack images in 3-μm steps were conducted for additional analysis using NIS-Elements software(Nikon, NIS-AR, Tokyo, Japan) and the ImageJ Fiji program (NIH, MD, USA). Any conversion in brightness or contrast was evenly applied to the entire image set. Confocal microscopic images were analyzed using the ImageJ program (NIH).

### 4.6. DAB Staining

The 30-μm-thick coronal sections for the striatum and SNpc were used for DAB staining. At first, the sections were washed three times in 0.3% TBS-T (0.3% Triton X-100 in TBS) for 5 min. Then, they were incubated in 1% H_2_O_2_ for 20 min while rocking slowly. After that, the tissues were washed three times in 0.3% TBS-T for 10 min, and the samples were incubated in a blocking solution (5% BSA in 0.3% TBS-T) at room temperature for 2 h. Then, the samples were immunostained with a mixture of rabbit anti-tyrosine hydroxylase (diluted in 1:500; Pel-freez, P40101-150, AR, USA) in a blocking solution at 4 °C on a rocker overnight. The samples in the mixture of primary antibody were reacted for one more hour at room temperature and then were washed three times (for 30 min the first time and for 10 min the second and the third). Then, the samples were incubated with a biotinylated secondary antibody (diluted in 1:500; Vector Labs, PK-6101, NJ, USA) mixture in a blocking solution for 2 h at room temperature, and subsequently with ABC reagent (Vector Labs, PK-6101, USA) for 30 min at room temperature. After the reaction, the samples were washed four times in TBS-T for 10 min. The tissues were dipped in the 1X DAB solution (Roche, #11718096001, Basel, Switzerland) for 20 s and then washed in 0.3% TBS-T and PBS to stop the DAB reaction, and the slices were mounted onto the silane-coated slide glass and completely dried. For dehydration, the slide glasses were washed for 5 min in deionized water. Then, the slides were immersed in 70% ethanol for 5 min, and then moved to 80%, 90%, and 100% ethanol for 2 min for each session. Lastly, the slides were immersed three times in xylene for 10 min. The slide glasses were covered with cover glasses using toluene mounting solution (Thermofisher, SP15-100, MA, USA). Finally, the bright field images of these samples were obtained with a bright field microscope (Olympus, BX50, Tokyo, Japan).

For densitometric analysis of the striatum area with DAB staining, coronal sections of the striatum were observed under a bright field microscope (Olympus, BX50, Japan) using 1.25× magnification. The optical density of the DAB-stained striatum was determined at an equivalent frame range, with optical density in the corpus callosum used as a reference.

### 4.7. Behavioral Tests

#### 4.7.1. Stepping Test

The stepping test was performed at three different time points: two weeks after AAV-rDDC-mCherry or AAV-rDDC-hM3Dq-mCherry injection (pre), and three weeks after AAV-CMV-A53T injection (PD), and after CNO injection (post). The stepping test was performed as previously described [19] with a slight modification. A habituation session was performed to familiarize rats with the treadmill the day before the test. At the test session, both hindlimbs and one forelimb were gently fixed by the experimenter, and another forelimb was allowed to step. One trial was for 10 s. All sessions were video-recorded to allow for the counting of the stepping numbers. Two trials were conducted in one test and the numbers of each forelimb step were averaged.

#### 4.7.2. Balance Beam Test

The balance beam test was performed at three different time points, following the stepping test. The balance beam test was performed as previously described [20] with a slight modification. In detail, the day before the first test, habituation was performed once. The habituation was conducted with a normal beam (2.5 cm in width, 1 m in length). A rat was placed on the top platform with home-cage bedding for three minutes. Then, we placed the rat on the bottom platform so that it can cross the beam from the bottom to the top platform. After this habituation, the main test was performed with a narrower beam (1 cm in width, 1 m in length) in the same way. This test was repeated twice at once. All the experiments were video-recorded to allow for counting the number of foot slips and evaluating the beam score. These counted numbers of foot slips were averaged in the two trials.

### 4.8. Statistical Analyses

Statistical analyses were performed using Prism 9 (GraphPad Software, Prism 9, CA, USA). For the confirmation of normal distribution, the Shapiro–Wilk normality test was assessed. Differences between the two different groups with normal distribution were analyzed with the two-tailed Student’s unpaired *t*-test. When the data are not normally distributed, the Mann–Whitney unpaired *t*-test was employed. For the comparison of multiple groups with normal distribution, one-way ANOVA with Tukey’s multiple comparison test was assessed. For assessment of the change of a group by a certain intervention, the significance of data was assessed by repeated measure one-way ANOVA with Tukey’s multiple comparison test. For the comparison of multiple groups with unnormal distribution, Friedman ANOVA with Dunn’s multiple comparison test was assessed. *p* < 0.05 was considered to indicate statistical significance throughout the study. The significance level is represented as asterisks (* *p* < 0.05, ** *p* < 0.01, *** *p* < 0.001, **** *p* < 0.0001; ns, not significant). All data are presented as mean ± SEM. No statistical method was used to predetermine the sample size. Sample sizes were determined empirically based on our previous experiences or the review of similar experiments in the literature. The numbers of animals used are described in the corresponding figure legends. All experiments were done with at least three biological replicates. Experimental groups were balanced in terms of animal age, sex, and weight.

## 5. Conclusions

In this study, we validated a DDC promoter in the rat brain. Utilizing this promoter, we demonstrated that the chemogenetic activation of the DDC^+^ neurons in the SNpc significantly alleviated the parkinsonian motor symptoms and the loss of nigrostriatal TH in the A53T PD model. Taken together, we expect our study stimulates future investigations on gene therapy targeting the dormant DA neurons for treating PD.

## Figures and Tables

**Figure 1 ijms-24-02491-f001:**
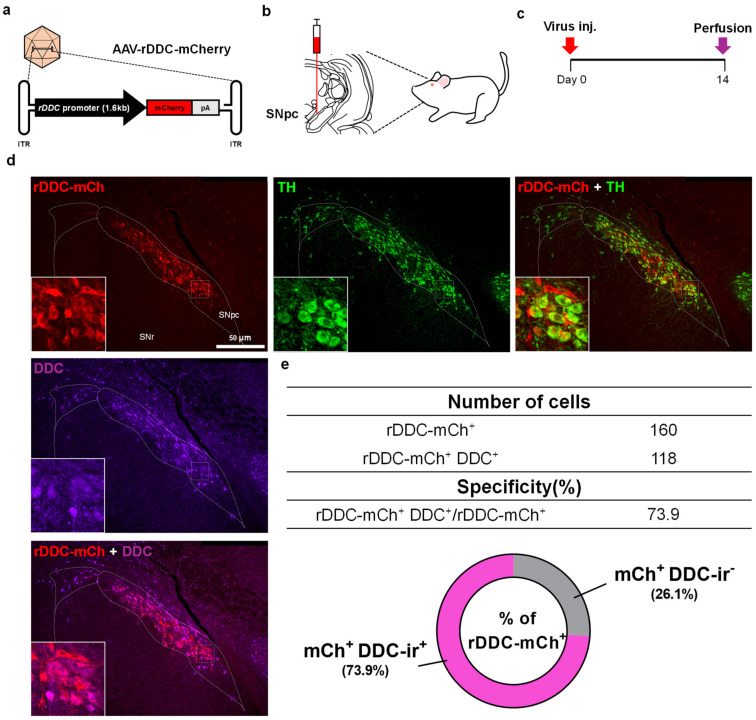
Development and specificity validation of rDDC-promoter. (**a**) Schematic diagram of AAV-rDDC-mCherry virus. Expression of the mCherry is regulated by an upstream rDDC-promoter. The polyadenylation signal (pA) marks the end of the transcriptional cassette. The entire cassette is flanked by ITRs that serve as signals/primers for vector DNA replication and encapsidation. The promoter sequence was referred to as NM_12545, chr14-:91,998,135-91,996,504 (assembly version rn6); (**b**) schematic diagram of stereotaxic virus injection into SNpc; (**c**) experimental timeline; (**d**) representative confocal images of SNpc, which was injected with AAV-rDDC-mCherry and immunolabeled with anti-TH and anti-DDC antibodies. Scale bar = 50 μm; (**e**) top, number of cells expressing mCherry and/or DDC. Bottom, quantification of the specificity of mCherry^+^/DDC^+^ neurons out of whole mCherry^+^ neurons in SNpc (*n* = 4 rats).

**Figure 2 ijms-24-02491-f002:**
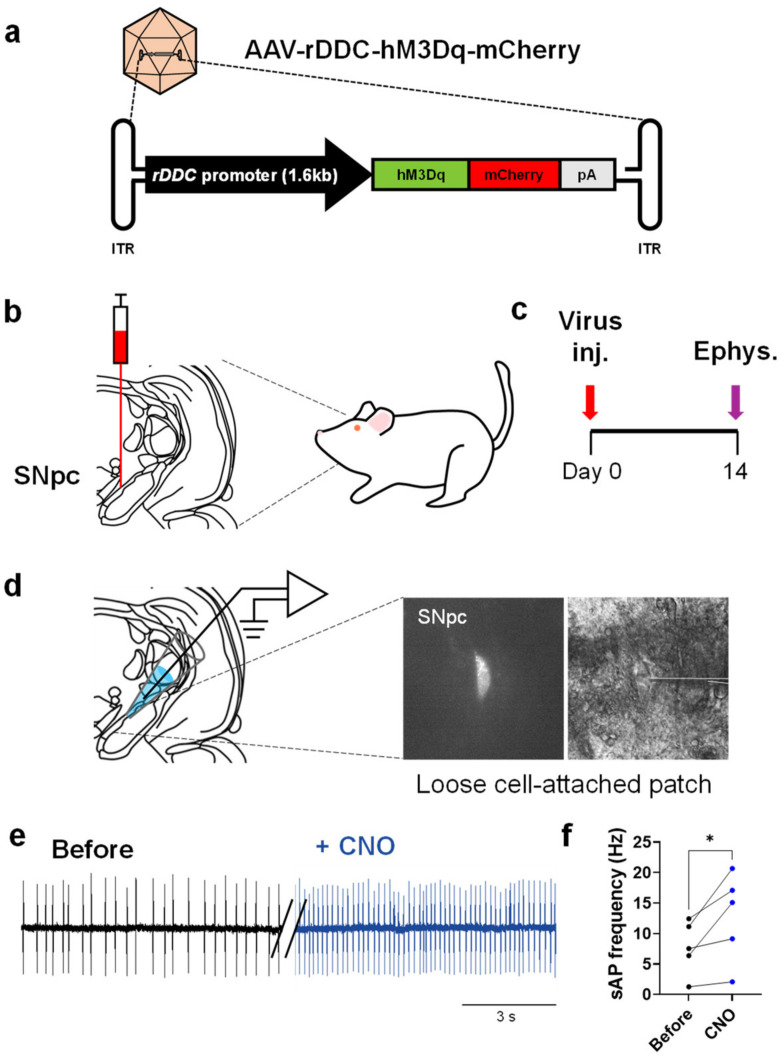
Gq-DREADD-mediated chemogenetic activation of DDC^+^ neurons increases the firing rate in SNpc. (**a**) The schematic for a viral vector that contains hM3Dq and mCherry; (**b**) schematic diagram of stereotaxic injection of AAV-rDDC-hM3Dq-mCherry virus; (**c**) experimental timeline; (**d**) representative images of the DDC^+^ neurons with loose cell-attached patch; (**e**) representative trace of spontaneous action potential firing of DDC^+^ neuron before and after CNO bath application (5 μM); (**f**) quantification of firing rate. The data normality was assessed by Shapiro–Wilk normality test. Statistical significance was calculated by two-tailed Student’s paired *t*-tests. The significance levels are represented as asterisks (* *p* < 0.05).

**Figure 3 ijms-24-02491-f003:**
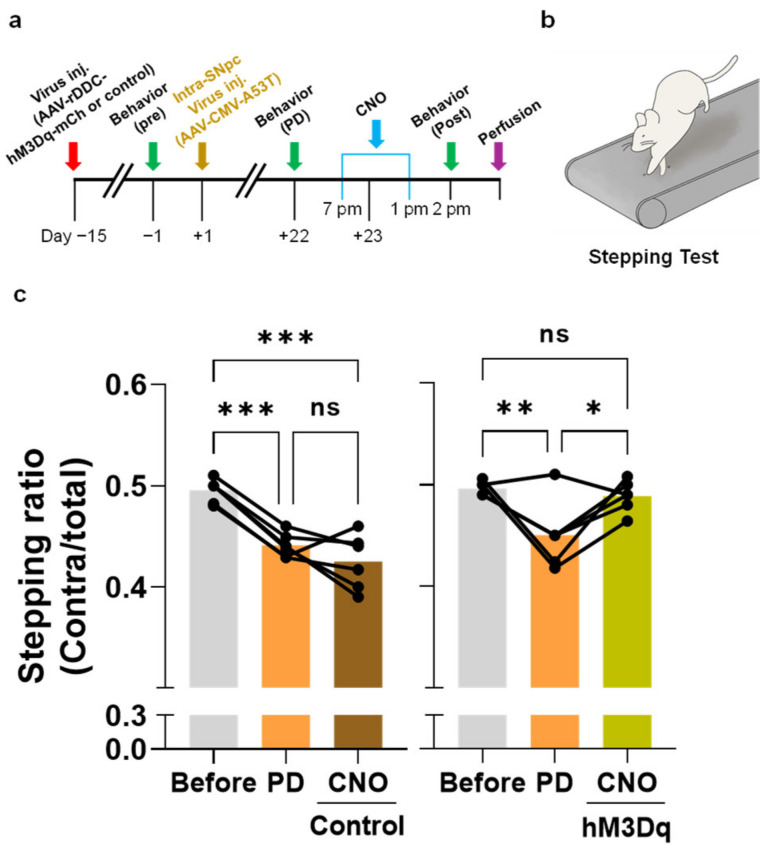
Chemogenetic activation of DDC neurons recovers PD-like motor symptoms of the A53T rat PD model in the stepping test. (**a**) Timeline of the experimental procedures; (**b**) schematic figure of stepping test; (**c**) quantification of stepping ratio (contralateral step number/total step number) before AAV-A53T virus injection (pre), 3 weeks after AAV-A53T virus injection (PD), and after intraperitoneal CNO (1.5 mg/kg) administration (post) (Control, *n* = 6 rats; hM3Dq, *n* = 6 rats). The data normality was assessed by Shapiro–Wilk normality test. Statistical significance was calculated by repeated-measure one-way analysis of variance (ANOVA) with Tukey’s multiple comparison test. The significance levels are represented as asterisks (* *p* < 0.05, ** *p* < 0.01, *** *p* < 0.001; ns, not significant).

**Figure 4 ijms-24-02491-f004:**
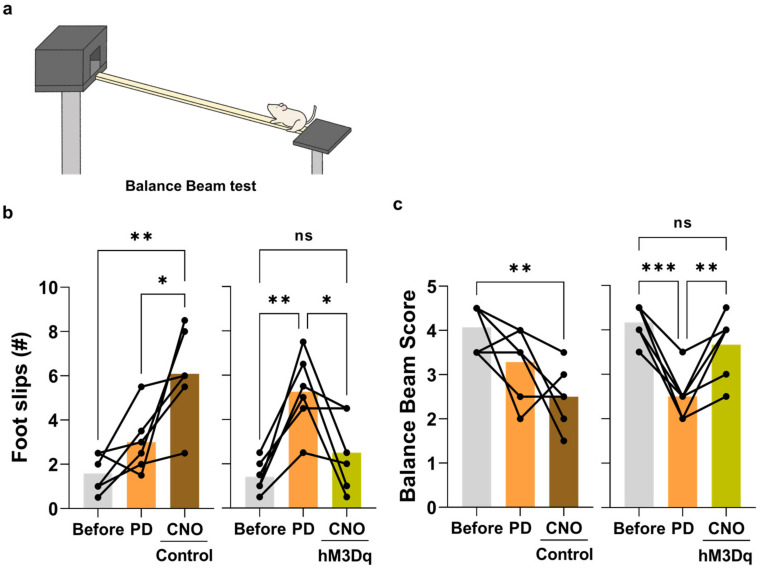
Chemogenetic activation of DDC neurons recovers PD-like motor symptoms of the A53T rat PD model in the balance beam test. (**a**) Schematic figure of balance beam test; (**b**) quantification of slippery step number at each time point (Control, *n* = 6; hM3Dq + CNO, *n* = 6); (**c**) assessment of balance beam score, which is classified by the following criterion: 1, the rat tried to traverse the beam, but fell; 2, the rat traversed the beam with many slips (more than five times); 3, the rat traversed the beam with several foot slips (three to five); 4, the rat traversed the beam with few foot slips (one to two); 5, the rat traversed the beam without any slips of the hindlimb. The data normality was assessed by Shapiro–Wilk normality test. When the data is normally distributed, the statistical significance was calculated by repeated-measure one-way analysis of variance (ANOVA) with Tukey’s multiple comparison test (b and c-right). When the data is not normally distributed and the statistical significance was calculated by Friedman analysis of variance (ANOVA) with Dunn’s multiple comparison test (c-left). The significance levels are represented as asterisks (* *p* < 0.05, ** *p* < 0.01, *** *p* < 0.001; ns, not significant).

**Figure 5 ijms-24-02491-f005:**
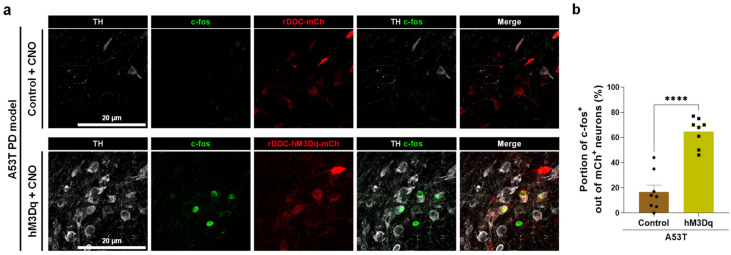
Chemogenetic activation of DDC^+^ neurons increases c-Fos expression in DAergic neurons of the A53T rat model. (**a**) Representative confocal images of SNpc, which were immunolabeled with anti-TH and anti-c-fos antibodies. Scale bar = 20 μm; (**b**) The proportion of c-fos^+^ neurons in Rat SNpc (Control, *n* = 8 rats; hM3Dq, *n* = 8 rats). The data are presented as the mean ± SEM. The data normality was assessed by Shapiro–Wilk normality test. Statistical significance was calculated by the two-tailed unpaired Student’s *t*-test. The significance levels are represented as asterisks (**** *p* < 0.0001).

**Figure 6 ijms-24-02491-f006:**
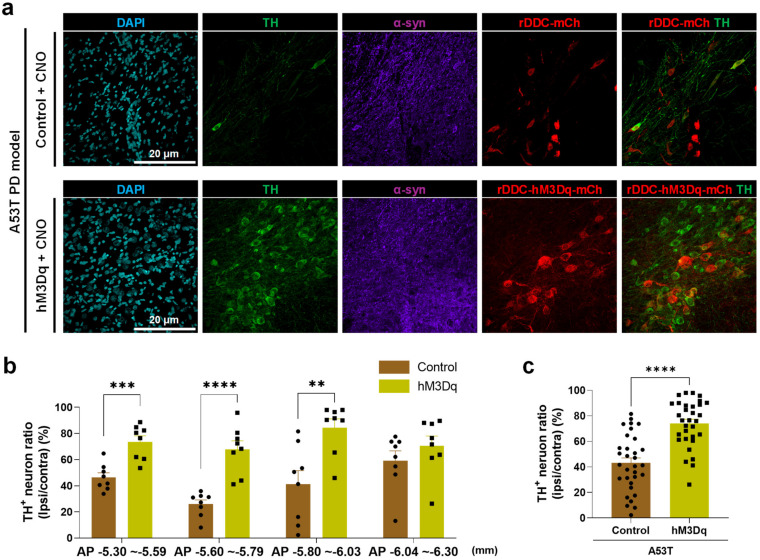
Chemogenetic activation of DDC-positive neurons alleviates the TH loss in SNpc of the A53T rat PD model. (**a**) Representative confocal images of SNpc, which were immunolabeled with anti-TH and anti-α-syn antibodies. Scale bar = 20 μm; (**b**) the ratio of remaining TH+ neurons in the ipsilateral SNpc compared to the contralateral side at four different coronal sections from rostral to caudal SNpc (Control, *n* = 8, hM3Dq, *n* = 8). The number of TH+ neurons in both groups and both hemispheres were counted. The X-axis indicates the anteroposterior (AP) coordinates from bregma; (**c**) bar graph which displays all data from each AP coordinates. The data are presented as the mean ± SEM. The data normality was assessed by Shapiro–Wilk normality test. Statistical significance was calculated by the two-tailed unpaired Student’s *t*-test. The significance levels are represented as asterisks (** *p* < 0.01, *** *p* < 0.001, **** *p* < 0.0001).

**Figure 7 ijms-24-02491-f007:**
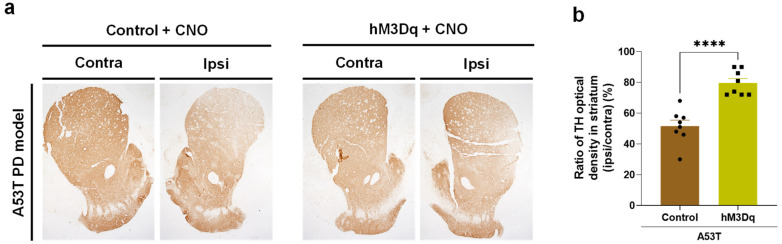
Chemogenetic activation of DDC^+^ neurons alleviates the striatal TH loss in the A53T rat PD model. (**a**) Representative images of striatum, which was immunolabeled with anti-TH antibody; (**b**) the optical density of TH in the ipsilateral striatum compared to the contralateral side (Control, *n* = 8 rats; hM3Dq, *n* = 8 rats). The data are presented as the mean ± SEM. The data normality was assessed by Shapiro–Wilk normality test. Since the data was not normally distributed, statistical significance was calculated by the Mann–Whitney test. The significance level is represented as asterisks (**** *p* < 0.0001).

## Data Availability

The data presented in this study are available in the Appendix A.

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
