# Peer review of "DDC-Promoter-Driven Chemogenetic Activation of SNpc Dopaminergic Neurons Alleviates Parkinsonian Motor Symptoms"

_ijms, 2023, doi:10.3390/ijms24032491_

Round 1

Reviewer 1 Report

Dear Authors,

Congratulations! Your paper is interesting, captivating, and brings novel, insightful possibilities.

Some modifications are needed:

In order to make it easier for the readers to visualize and understand your article,

The "Materials and Methods" chapter should be the second chapter in your manuscript 

The "Discussions" chapter should be the third chapter in your manuscript 

The "Conclusions" chapter should be the fourth chapter in your manuscript 

Also, you can split the figures in order to make the images wider, to be better seen.

Furthermore, it would be fantastic if you could include another chapter in which you could explain the relevance of your findings in an animal model as well as in humans.

Moreover, you could try to find more articles that would discuss a converging theme and extrapolate your results and the applicability of your results in comparison.

Author Response

**Please find the attached file for better readability.**

Congratulations! Your paper is interesting, captivating, and brings novel, insightful possibilities.

  • We appreciate the reviewer for the positive responses and the constructive comments. We added our response to every single comment in a point-by-point manner. All revised text is marked in blue in the manuscript file.

1. Some modifications are needed: In order to make it easier for the readers to visualize and understand your article, The "Materials and Methods" chapter should be the second chapter in your manuscript. The "Discussions" chapter should be the third chapter in your manuscript. The "Conclusions" chapter should be the fourth chapter in your manuscript.

  • We agree with the reviewer’s comments. As the reviewer suggested, we rearranged the order of chapters and additionally prepared “Conclusions” as the fourth chapter in our manuscript.

2. Also, you can split the figures in order to make the images wider, to be better seen.

  • We understand the reviewer’s comment. In response to the reviewer’s comment, we divided the figure 3 into figure 3 and 4.
  • In addition, to increase the readability, we increase the size of Figure 1d & 1e in the revised manuscript.

3. Furthermore, it would be fantastic if you could include another chapter in which you could explain the relevance of your findings in an animal model as well as in humans.

  • We appreciate the reviewer for constructive comment. In response to the reviewer’s comment, we additionally discuss the possible clinical relevance in human patients in the Discussion chapter of the revised manuscript (see Discussion 4.3).

4. Moreover, you could try to find more articles that would discuss a converging theme and extrapolate your results and the applicability of your results in comparison.

  • In response to the reviewer’s comment, we additionally discussed a converging theme of the neuromodulation therapies for PD and extrapolate our results by adding a paragraph with 14 more references (see Discussion 4.2 of the revised manuscript).

Reviewer 2 Report

The study by Dong-Chang Seo et al presents some exciting data and hypotheses regarding dormant DDC+ neurons and their potential role if modulated properly, in partial/full recovery in animal A53T mode of PD. Even though the research offers very compelling evidence that chemogenetic activation of DDC neurons may alleviate PD-associated symptoms, there are some points that need to be addressed:

1) Can Authors support with literature the statement that DDC neurons are ``dormant``? What is the meaning of the word dormant in this context?

2) The canonical biosynthesis pathway of dopamine encompasses both tyrosine hydroxylase (TH) converting L-tyrosine to L-DOPA, which is a substrate for DOPA-decarboxylase (DDC) which converts L-DOPA to dopamine. If the Reviewer understood correctly, you are suggesting that there is a subpopulation of DDC+/TH- neurons.  Figure 1 also shows a similar pattern of TH immunofluorescence and AAV-rDDC-mCherry+ neurons which implies that all neurons are TH+ and DDC+. Did you perform colocalization of DDC and TH in the healthy (contralateral) hemisphere?

3) When did you assess the c-FOS immunoreactivity? It is very unusual and highly unlikely that acute application of CNO rescues TH+ neurons in such manner. Can the Authors show the analysis of TH loss in the A53T induced PD model? 

4) This Reviewer is not convinced that ``dormant`` neurons may regain TH expression, given that TH expression is an indirect measure of loss of dopaminergic neurons in SNpc and not of transient downregulation which can be upregulated with DREADD ligand. 

5) The behavioral tests are not described in a manner that may allow repeatability and reproducibility. Also, the number of animals is not big enough to ensure the validity of the phenomenon. Did you perform the statistical tests checking for the normality of your data? Given the small number of animals per group for behavioral analysis and evident deviation it is possible that you do not have a normal distribution. Furthermore, there are, I presume, some errors in the M&M section, i.e. 10m/s for the speed of the treadmill (its probably 10cm/s, and its states 2 mg/kg of CNO, while in the Figure legends, it states 1.5 mg/kg)

6) To fully verify the effects of the treatment it would be convenient to perform an HPLC analysis of dopamine levels in the striatum.

7) Can the Author provide an analysis of the expression (immunochemistry od western blot) of alpha-synuclein expression following the viral infection?

8) Why Authtors did not have control for A53T viral injection?

9) Discussion must be improved and more literature data included 

Author Response

Reviewer 2

The study by Dong-Chan Seo et al presents some exciting data and hypotheses regarding dormant DDC+ neurons and their potential role if modulated properly, in partial/full recovery in animal A53T mode of PD. Even though the research offers very compelling evidence that chemogenetic activation of DDC neurons may alleviate PD-associated symptoms, there are some points that need to be addressed:

  • We appreciate the reviewer for the insightful summary and the constructive comments. We added our response to every single comment in a point-by-point manner. All revised text is marked in blue in the manuscript file.

1. Can Authors support with literature the statement that DDC neurons are ``dormant``? What is the meaning of the word dormant in this context?

  • We previously reported that the aberrant astrocytic GABA tonically inhibits DAergic neuronal activity in the SNpc, leading to substantial TH loss which can be restored by optogenetic activation (Heo et al., Current Biology, 2020). In the same paper, we further demonstrated the existence of TH-negative but DDC-positive neurons in the SNpc of three different PD rodent models including MPTP, 6-OHDA, and A53T models as well as post-mortem SNpc tissues of PD patients. This population of neurons is thought to be under the process of neurodegeneration. However, since they are still alive and their TH expression is recoverable, we named it “dormant” DA neurons.
  • In response to the reviewer’s comment, we additionally discussed the meaning of “dormant” neurons in the Discussion section 4.1 of the revised manuscript.

2. The canonical biosynthesis pathway of dopamine encompasses both tyrosine hydroxylase (TH) converting L-tyrosine to L-DOPA, which is a substrate for DOPA-decarboxylase (DDC) which converts L-DOPA to dopamine. If the Reviewer understood correctly, you are suggesting that there is a subpopulation of DDC+/TH- neurons. Figure 1 also shows a similar pattern of TH immunofluorescence and AAV-rDDC-mCherry+ neurons which implies that all neurons are TH+ and DDC+. Did you perform colocalization of DDC and TH in the healthy (contralateral) hemisphere?

  • We understand the reviewer’s concern. As the reviewer correctly pointed out, almost all neurons express both TH and DDC in the physiological condition. On the contrary, in the SNpc of PD mouse model, we previously demonstrated that the population of DDC-positive/TH-negative neurons (Heo et al., Current Biology, 2020). Therefore, we did not perform co-localization assessment of DDC and TH in the contralateral hemisphere.

3. When did you assess the c-FOS immunoreactivity? It is very unusual and highly unlikely that acute application of CNO rescues TH+ neurons in such manner. Can the Authors show the analysis of TH loss in the A53T induced PD model?

  • We performed cardiac perfusion with paraformaldehyde 1.5-to-2 hours after the last CNO administration (as depicted in the Figure 3a of the original version of manuscript). Previous studies have also shown that CNO-mediated hM3Dq activation causes Ca2+ rise and increases c-Fos immunoreactivity after 2 hours (Zhan et al., JoVE, 2019; Luo et al., Front Mol Neurosci., 2018). In response to the reviewer’s comment, we added a description of the timing of tissue fixation by cardiac perfusion in the Methods section of the revised manuscript, as shown below:
    • “Cardiac perfusion was performed under deep anesthesia by 2% avertin with nor-mal saline and ice-chilled 4% PFA solution immediately after the last behavior test finished (1.5 to 2 hours after the last CNO administration).”
  • The TH expression has been known to be regulated by neuronal activity (Joh et al., PNAS, 1978; Kilbourne et al., JBC, 1992; Aumann et al., JNC, 2011). In particular, Aumann et al. directly stated in the title that “Neuronal activity regulates expression of tyrosine hydroxylase in adult mouse substantia nigra pars compacta neurons.” As mentioned above, we also previously demonstrated that 1-day optogenetic activation of SNpc neurons in two different PD rodent models significantly rescued the TH expression (Heo et al., Current Biology, 2020). Taken together, TH expression could be recovered when the neuron is still alive. In response to the reviewer’s comment, we additionally discussed the possibility of activity-dependent TH regulation in the Discussion section of the revised manuscript.
    • “Previous studies have also strongly supported the idea that TH expression is regulated upon neuronal activity [25-27].”
  • In Figure 5 and 6 of the original manuscript (Fig 6 and 7 of the revised manuscript), we showed the SNpc TH+ cell counting analysis (Fig. 6b-c) and the optical density analysis of striatal TH expression (Fig. 7b) in the ipsilateral SNpc, compared to the contralateral side, in the A53T-induced PD model (see below).
  • We sincerely hope that our description satisfies the reviewer.

4. This Reviewer is not convinced that ``dormant`` neurons may regain TH expression, given that TH expression is an indirect measure of loss of dopaminergic neurons in SNpc and not of transient downregulation which can be upregulated with DREADD ligand.

  • As described above, TH expression can be regulated by neuronal activity (Joh et al., PNAS, 1978; Kilbourne et al., JBC, 1992; Aumann et al., JNC, 2011). In particular, Aumann et al. described the possible mechanism of activity-dependent TH regulation.
    • “The first is by intracellular Ca2+ influencing TH transcription via Ca2+-dependent phosphorylation (e.g. protein kinase C or calcium/calmodulin-dependent protein kinases) and de-phosphorylation (e.g. calcineurin) of transcription factors (e.g. cAMP-response element binding protein or c-Fos). The rodent (rat and mouse) TH promoter contains cAMP (~40 bp and ~90 bp upstream), activator protein 1 (~200 bp upstream) and calcium response element sequences upstream of the transcription start site (Kumer and Vrana 1996; Sabban 1997; Sabban and Kvetnansky 2001). The TH gene may also undergo Ca2+-dependent epigenetic modifications (e.g. Flavell and Greenberg 2008). Intracellular Ca2+ influences gene expression in various cell-types leading, in some cases, to remarkable cellular phenotypic changes (e.g. Flavell and Greenberg 2008; Dolmetsch et al. 1997; Matza and Flavell 2009) and neurons are the most exquisite Ca2+ pumps in the body.” (from Aumann et al., JNC, 2011)
  • Therefore, hM3Dq-mediated chemogenetic activation of the neurons elicits the intracellular Ca2+ rise, which is very likely to contribute to the TH regulation. In addition, we previously reported that the TH expression could be restored by ChR2-mediated optogenetic stimulation of the SNpc neurons in the rodent PD models.
  • In summary, TH expression is not only an indirect measure of loss of dopaminergic neurons, but it can be transiently regulated upon neuronal activity.

5. The behavioral tests are not described in a manner that may allow repeatability and reproducibility. Also, the number of animals is not big enough to ensure the validity of the phenomenon. Did you perform the statistical tests checking for the normality of your data? Given the small number of animals per group for behavioral analysis and evident deviation it is possible that you do not have a normal distribution.

  • We appreciate the reviewer’s comment. As the reviewer pointed out, relatively small N number does not ensure the normal distribution of the data points. Therefore, in response to the reviewer’s comment, we estimated the normality of the data distribution by Shapiro-Wilk test using Prism 9. When the data is not normally distributed (Figure 4C left and 7B), we assessed the significance with non-parametric test (Friedman ANOVA with Dunn’s multiple comparison test for 4C; Mann-Whitney test for 7B) instead of parametric test (RM one-way ANOVA with Tukey’s multiple comparison test for 4C; two-tailed Student’s unpaired t-test for 7B).
  • In the revised manuscript, we changed the figure 4C with the figure legends and M&M section.

6. Furthermore, there are, I presume, some errors in the M&M section, i.e. 10m/s for the speed of the treadmill (its probably 10cm/s, and its states 2 mg/kg of CNO, while in the Figure legends, it states 1.5 mg/kg)

  • We regret our unprofessional mistakes and appreciate the reviewer’s comment. In response to the reviewer's comment, we revised them from 10 m/sec to 10 cm/sec, and from 2 mg/kg of CNO to 1.5 mg/kg of CNO.

7. To fully verify the effects of the treatment it would be convenient to perform an HPLC analysis of dopamine levels in the striatum.

  • Even though we also think that measuring striatal dopamine levels by HPLC would be great for verifying the effects of the chemogenetic approach, there are some practical limitations to do all experiments we want to conduct. We spare this topic for our follow-up study. We deeply appreciate for the reviewer’s comment.
  • Nonetheless, since TH is the key enzyme for dopamine synthesis, TH expression is highly correlated with dopamine level in the striatum. Therefore, we expect that the results of dopamine levels assessed by HPLC would consistent with our TH staining results.

8. Can the Author provide an analysis of the expression (immunochemistry or western blot) of alpha-synuclein expression following the viral infection?

  • Yes, we displayed the alpha-synuclein staining results in Figure 5a of the original manuscript (Fig 6a of the revised manuscript). Alpha-synuclein is well expressed in the SNpc upon AAV-A53T viral injection.

9. Why Authors did not have control for A53T viral injection?

  • As the reviewer pointed out, having more control makes the finding more convincing. However, this study focuses on the effect of chemogenetic manipulation rather than the effect of A53T viral injection. Therefore, to ethically reduce the animal consumption, we had an essential control group for hM3Dq-mediated chemogenetic activation. We humbly request the reviewer’s generous understanding.

10. Discussion must be improved and more literature data included

  • In response to the reviewer’s insightful comment, we additionally discussed on 1) the necessity of DDC promoter validation, 2) recent studies on neuromodulation strategy for PD treatment, and 3) Advantages of AAV-mediated gene therapy. We also added 22 more references. Please find the revised Discussion section of the new version of the manuscript file.

Round 2

Reviewer 2 Report

The Authors have answered all of the Reviewer's concerns and improved the quality of the manuscript.